# Limited Stress Response to Transplantation in the Mediterranean Macroalga *Ericaria amentacea*, a Key Species for Marine Forest Restoration

**DOI:** 10.3390/ijerph191912253

**Published:** 2022-09-27

**Authors:** Silvia Chemello, Geraldina Signa, Antonio Mazzola, Tania Ribeiro Pereira, Isabel Sousa Pinto, Salvatrice Vizzini

**Affiliations:** 1Department of Earth and Marine Sciences (DiSTeM), University of Palermo, 90123 Palermo, Italy; 2Interdisciplinary Centre of Marine and Environmental Research (CIIMAR), University of Porto, 4450-208 Matosinhos, Portugal; 3Consorzio Nazionale Interuniversitario per le Scienze del Mare (CoNISMa), 00196 Rome, Italy

**Keywords:** coastal restoration, habitat conservation, reforestation, *Cystoseira*, acclimation, fatty acids, phenolic compounds, lipids

## Abstract

In the Mediterranean Sea, brown macroalgae represent the dominant species in intertidal and subtidal habitats. Despite conservation efforts, these canopy-forming species showed a dramatic decline, highlighting the urge for active intervention to regenerate self-sustaining populations. For this reason, the restoration of macroalgae forests through transplantation has been recognized as a promising approach. However, the potential stress caused by the handling of thalli has never been assessed. Here, we used a manipulative approach to assess the transplant-induced stress in the Mediterranean *Ericaria amentacea*, through the analysis of biochemical proxies, i.e., phenolic compounds, lipids, and fatty acids in both transplanted and natural macroalgae over time. The results showed that seasonal environmental variability had an important effect on the biochemical composition of macroalgae, suggesting the occurrence of acclimation responses to summer increased temperature and light irradiance. Transplant-induced stress appears to have only amplified the biochemical response, probably due to increased sensitivity of the macroalgae already subjected to mechanical and osmotic stress (e.g., handling, wounding, desiccation). The ability of *E. amentacea* to cope with both environmental and transplant-induced stress highlights the high plasticity of the species studied, as well as the suitability of transplantation of adult thalli to restore *E. amentacea* beds.

## 1. Introduction

Large brown macroalgae (mostly represented by fucoid species) form dense canopies or even extensive forests along the Mediterranean intertidal and subtidal rocky shores, where they play a key ecological role. Most of these canopy-forming species have, indeed, a crucial functional role as “habitat engineers” of the coastal landscape [1], representing nursery areas for reef fish [2], sustaining complex food webs and maintaining high levels of associated biodiversity [3,4]. However, in the last decade, brown macroalgae species such as *Cystoseira sensu latu* (including the genera *Cystoseira*, *Ericaria* and *Gongolaria* [5]) or *Sargassum* genus (e.g., *Sargassum hornschuchii* and *S. vulgare*) are experiencing a dramatic decline in abundance and distribution in many Mediterranean locations, i.e., [6,7], also compromising the ecosystem services they provide, and their associated goods and benefits for humans [8]. This is especially true in urbanized coastal areas, due to the synergistic effect of anthropogenic local disturbances and climate stress [7,9,10,11].

Despite substantial conservation efforts (most of these species are protected by the Barcelona Convention, 1972, and the Bern Convention, 1976), many of the depleted populations have not been able to recover naturally due to very limited dispersal ability [12], highlighting the need for active intervention measures. For this reason, restoration of several macroalgal forests has recently been attempted using different approaches including transplanting of adults taken from healthy forests and ex situ or in situ recruitment enhancement (i.e., germling outplanting), recently reviewed by Cebrian et al. [13]. However, macroalgal restoration is still at an “innovation phase” and, despite significant efforts recently made in the framework of EU projects (e.g., AFRIMED, MERCES, ROCPOP-life), it has received little attention compared to other coastal habitats (e.g., seagrasses and saltmarshes) [14]. One of the main bottlenecks of macroalgal restoration is the high mortality rate during the early settlement stages [15,16], but it is still unclear which biotic or abiotic factors contribute to this initial demise [1]. Although several post-settlement factors, such as genetic, environmental and “consumptive” factors (e.g., grazing), play critical roles in shaping restoration success (see [10]), mechanical stress caused by handling and clipping of macroalgae (both germlings and adults) is likely to be one of the main factors responsible for the low survival rate in the early settlement stage. For example, *Cystoseira compressa* thalli displayed evident stress after transplanting, consisting of loss of primary branches and a less flourishing and more prostrate appearance [17]. However, transplant stress has rarely been evaluated, except by assessing mortality rates. As reviewed by Bellgrove et al. [1], manipulative studies are urgently needed to untangle the limitations and maximize the effectiveness of restoration efforts on intertidal rocky shores.

It is acknowledged that macroalgae have developed acclimation and adaptation mechanisms, as well as defence strategies to cope with external stressors, mainly based on biochemical and physiological adaptations to both environmental and biotic stress [18]. Among these, the most common strategies developed by macroalgae to prevent or reduce stress caused by the exposure to different stressors are (i) the production and accumulation of secondary metabolites that play a role in photoprotection and defence against grazers and epiphyte cover [19,20]; (ii) the activation of antioxidant metabolism against oxidative stress, such as ultraviolet (UV) radiation [21,22], metal exposure [23,24], variation in salinity, temperature, light, nutrients, and desiccation, are associated with the formation and accumulation in the seaweed cells of reactive oxygen species (ROS) that cause oxidative stress [25,26,27].

Two of the most studied metabolic responses associated with either environmental, biotic or both stresses, are the production of phenolic compounds and the change in lipid concentration and fatty acid composition. In more detail, macroalgae are rich in phenolic compounds, which are mainly known to be involved in protection mechanisms against abiotic stresses (e.g., photoprotection against UV irradiance) [28,29,30]. Anyway, phenols show high plasticity, responding to other kinds of factors, such as nutrient levels, temperature, salinity, grazing and mechanical wounding [31,32,33,34]. Macroalgae are also naturally rich in fatty acids (FAs), which are a major component of lipids with important antimicrobial and deterrent properties [20,35]. Most of the FAs produced by macroalgae are polyunsaturated fatty acids (PUFAs) that are key components of the cell membrane phospholipids [18]. Adjustments in lipid content and FA composition and relative abundance have been observed in response to changes in environmental variables [36,37,38,39] as one of the first adaptive responses to environmental stressors involving adjustment in the fluidity of the cell membranes [40].

Using biochemical markers to evaluate the transplant stress response is a novel and promising approach, which, to our knowledge, has barely been used. FA analysis was indeed used to assess the adaptive response of corals to transplantation [41]. For macroalgae, other variables, such as changes in photosynthetic performance or associated microbiota, have been assessed in transplanted thalli, with the main aim of determining the effects of environmental changes after displacement [21,42].

The main goal of this study was to assess the stress response to transplantation of the macroalga *Ericaria amentacea*, (ex *Cystoseira amentacea*), a Mediterranean endemic fucoid species that colonize the upper intertidal zone forming dense monospecific belts. Hypothesizing that the change in environmental conditions and the potential physical damage due to clipping, handling and displacement of thalli may stress the macroalgae triggering a biochemical response; the variation in the concentration of phenolic compounds and the fatty acid profiles and classes was assessed to identify subtle effects of the transplantation that may lead to low survival rates of the transplanted thalli and compromise the effectiveness and success of marine macroalgal restoration and recovery actions.

## 2. Materials and Methods

### 2.1. Study Area and Experimental Setup

The experiment was carried out along the Northern coast of Sicily (Italy), within the “Capo Gallo-Isola delle Femmine” marine protected area (38°12′37.4″ N, 13°17′11.8″ E). Two sampling sites, hereafter called Barcarello and Capo Gallo (Figure 1), were selected along the coast as both are characterized by the presence of natural and extensive *Ericaria amentacea* beds growing on natural rocky shores, as well as comparable substrate, orientation, light and hydrodynamic conditions. At both sites, *E. amentacea* occupies the upper intertidal zone, in strict association with calcareous algae (*Lithophyllum lichenoides* Philippi, 1837) and biogenic structures formed by the vermetid mollusc—*Dendropoma cristatum* Biondi, 1859 [43,44].

The transplantation was carried out in late May 2018 and was followed for about four and a half months (up to September 2018). The transplantation protocol proposed by [45,46] consisted of four phases: (i) at each site, 20 random plots (20 × 20 cm), interspersed inside the natural *E. amentacea* belt, were cleared by removing the macroalgal canopy; (ii) individual thalli of similar size with their holdfast still attached to the substrate, were collected from the nearby natural population (160 thalli per site, 320 in total) using a chisel; (iii) 8 holes with a diameter of 18 mm were drilled at each plot, and (iv) 8 thalli were transplanted at each plot using epoxy putty (Stucchi Veneziani) to fix the thalli to the substrate [45,46]. Before transplantation, all the thalli collected from the nearby donor patches were washed with freshwater to remove the associated epifauna. The transplantation phase was carried out with adequate hydrodynamic conditions since the epoxy putty takes 24 h to catalyse and the fixation of the thalli to the substratum is a fundamental step for the success of transplantation. Due to the long time and physical effort required for drilling the substratum, as well as the long catalysation time of the epoxy putty, simulating the natural density of *E. amentacea* was not feasible, since in the Mediterranean the macroalga can reach high densities, with a canopy cover between 80 and 100% [11].

After transplanting (T0) and at four subsequent time points (T1 after two weeks, and T2, T3 and T4 after a further one, two and three months, respectively), five thalli were randomly collected from each transplanted plot and nearby natural plots, as a reference. Water temperature and salinity were also measured at each time and site using a multiparametric probe (Hanna Instruments model HI98194). Following the collection, both transplanted and natural thalli were carefully placed in separated plastic bags and kept under dark and cold conditions until the arrival at the laboratory, where they were rinsed in deionized water, and then stored frozen at −80 °C. Frozen samples were then freeze-dried and ground into fine powder until further analysis.

### 2.2. Biochemical Analysis

#### 2.2.1. Total Phenolic Compounds

Analysis of the content of total phenolic compounds (TPCs) was carried out both on transplanted and natural thalli, following a modified protocol from [47,48]. TPC extraction was carried out by adding 80% MeOH to samples, followed by incubation for 24 h in dark and cold conditions. After centrifugation, the supernatant fraction was collected, and a colorimetric determination of TPCs was performed after reaction with 20% Na_2_CO_3_ and Folin–Ciocalteu reagents. Reaction time was about 2 h, with constant stirring at room temperature, and then extracted samples were read with the spectrophotometer at 765 nm. Results were expressed as mg g^−1^ of dry weight (dw).

#### 2.2.2. Lipids and Fatty Acids

Analysis of fatty acids (FAs) was carried out on both transplanted and natural thalli. Lipid extraction was performed on 100 mg of each sample, following a modified version of the Bligh and Dyer method [49], and using a MilliQ distilled water: methanol: chloroform mixture (1:2:1 *v*:*v*:*v*) with 0.01% BHT (butylated hydroxytoluene) to prevent lipid oxidation. Samples were then sonicated to improve lipid extraction and then centrifuged twice to separate the lipid phase from the aqueous phase. The lipid extracts were evaporated to dryness under a gentle nitrogen stream, weighed, and expressed in mg g^−1^ dw. Then, lipid extracts were subjected to acid-catalysed transesterification using methanolic hydrogen chloride to obtain fatty acid methyl esters (FAMEs), which were analysed by a gas chromatograph (GC-2010, Shimadzu Corporation, Kyoto, Japan) equipped with a BPX-70 capillary column (30 m length; 0.25 mm ID; 0.25 μm film thickness, SGE Analytical Science), and detected by a flame ionization detector (FID). Peaks were identified by retention times from mixed commercial standards (37FAME from Supelco; QUALFISH from Larodan). Tridecanoic and tricosanoic acids (C13:0 and C23:0) were used as surrogate standards, while pentacosanoic acid methyl ester (ME C25:0) was used as the internal standard for FAME quantification. Individual FAs data were expressed as a percentage of total FAs.

#### 2.2.3. Data Elaboration and Statistical Analysis

Differences in TPCs and lipid concentration, as well as in FA relative abundance and classes between treatments (factor treatment: fixed, 2 levels, transplanted and natural) over time (factor time: fixed, 5 levels, T0, T1, T2, T3, T4) in the two sites (factor site: random, 2 levels, Barcarello and Capo Gallo), were tested using permutational analysis of variance (PERMANOVA, 9999 permutations) using PRIMER v6.1.10 and PERMANOVA+ β20 software (Plymouth, UK). All factors were considered orthogonal. Pair-wise tests were used to check for significant post hoc differences. All the datasets were previously reembled using a Euclidean similarity distance matrix based on untransformed data for TPCs and lipid concentration, and on transformed data using the arcsine function for FAs.

Principal component analysis (PCA) was performed on the FA data to graphically highlight the differences found by PERMANOVA, superimposing the vectors of the FA molecules contributing most to the ordination (Pearson correlation > 0.6). Furthermore, analysis of similarity percentage (SIMPER) was carried out on untransformed FA data to identify which individual FAs contributed most to the dissimilarity observed, consistent with PERMANOVA results.

As regards the thalli transplanted at the Capo Gallo site, those collected at T3 did not present enough biomass to carry out all the analyses and consequently, they were not analysed for FAs. Moreover, during the last month (between T3 and T4), all the individuals transplanted were lost, likely because of the strong hydrodynamic conditions and the grazing pressure of the *Sarpa salpa* fish (author’s personal observations), therefore the last sampling date (T4) of Capo Gallo was not included in the statistical analysis.

## 3. Results

### 3.1. Environmental Variables

Throughout the experiment, surface water temperature varied from 17.4 °C at T0 (May) to approximately 28 °C at T3 (August) at both sites, followed by a decrease of up to about 25 °C at T4 (September). Salinity varied from 37.3 PSU at T0 in both sites to about 38.0 PSU at T4 (September) (Figure 2).

### 3.2. Total Phenolic Compounds

PERMANOVA highlighted significant differences in the total phenolic compounds (TPCs) for the interaction between time and site factors (MS = 19.3; Pseudo-F_(3,79)_ = 6.7; *p* ≤ 0.001): TPCs were lower at T4 than at other sampling times at the Barcarello site, while a temporal decrease in TPC content was evident at the Capo Gallo site (pairwise tests; Figure 3, Appendix A). Moreover, significant differences, at borderline level, resulted for the treatment factor (MS = 27.2; Pseudo-F(_1,79)_ = 133.6; *p* = 0.053) with higher values in the transplanted than natural macroalgae.

### 3.3. Lipids and Fatty Acids

The concentration of total lipids (TL) in *Ericaria amentacea* thalli decreased over time in both treatments and sites. Moreover, PERMANOVA revealed significant differences for the interaction between the treatment and site factors (MS = 118.3; Pseudo-F_(1,68)_ = 4.27; *p* < 0.05) with transplanted thalli showing lower TL concentration than natural thalli at both sites (Barcarello: *p* < 0.05; Capo Gallo: *p* < 0.01) (Appendix A).

Fatty acid (FA) analysis performed on natural and transplanted *E. amentacea* thalli identified 33 individual FAs at the Barcarello site and 32 at Capo Gallo. Overall, the dominant FAs were 16:0 (palmitic acid, PALM) among the saturated FAs (SFA), followed by 20:4 n6 (arachidonic acid, ARA) and 18:3 n3 (linolenic acid, ALA) among the polyunsaturated FAs (PUFA), and 18:1 n9 (oleic acid, OLE) among the monounsaturated FAs (MUFA) (Table 1 and Table 2). When comparing the FA profiles between treatments over time and sites, significant differences arose in the interactions between all three factors (PERMANOVA, treatment × time × site: MS = 0.002; Pseudo-F_(1,67)_ = 3.31; *p* < 0.05, Appendix A). Pair-wise tests showed that (i) both natural and transplanted macroalgae were different between all times at both sites (Appendix A); (ii) at the Barcarello site, the FA profile of natural and transplanted thalli did not differ at the two first sampling times (T1 and T2), but only at T3 and T4 (Appendix A); (iii) at the Capo Gallo site, the FA profile of natural thalli differed from that of transplanted ones at both T1 and T2 (Appendix A) (comparison between treatments was not performed at T3 and T4 because of the lack of samples).

The analysis of the principal components (PCA) of the fatty acid profiles explained almost 80% of the variation, with the first axis accounting for the greatest part (62.1%) of the explained variation, and the second axis accounting only for 16.6% (Figure 4). Within this framework, the ordination revealed that time represented the main driver of the sample distribution within the ordination, with samples from T1 to T4 moving gradually from the right to the left side of the plot, while samples collected at the beginning of the experiment (T0) clustered in the upper central part of the plot. As evident from the vectors superimposed to the ordination, samples collected at T0 were characterized by a high relative abundance of 20:4 n6, and T1 samples clustered on the right side of the graph because of the high relative abundance of ω-3 FAs (i.e., 18:3 n3-ALA, 18:4 n3-SDA, 20:4 n3 and 20:5 n3-EPA). On the other side, SFAs (i.e., 14:0, 15:0, 16:0-PALM and 18:0), 18:1 n9-OLE, 18:1 n7 and 20:2 n6 characterized most samples retrieved at T2, T3 and T4. Within each cluster driven by sampling time, the differences between sites and treatment are also evident.

SIMPER analysis revealed that the increase in 18:4 n3 (stearidonic acid, SDA) and 20:5 n3 (eicosapentaenoic acid, EPA) coupled with the decrease in 20:4 n6 (ARA) contributed most to the dissimilarity between T0 and T1 for both natural and transplanted macroalgae at both sites (Table 3). The only exception was due to the thalli transplanted at Capo Gallo, where 16:0 (PALM) contributed more than EPA (10.7 vs. 8.2%). After T0, SDA and EPA started to decrease, contributing to the differences found from T1 to T2 in both treatments and sites, together with an increase in PALM and 18:1 n9 (OLE). Only the natural macroalgae from the Capo Gallo site differed from this trend, showing a rather relevant increase in ARA. In the subsequent times, the FA profiles of the macroalgae collected from Barcarello changed differently in the two treatments. A greater increase in PALM and decrease in EPA was observed in the transplanted algae causing most of the dissimilarity between T2 and T3. Similarly, the subsequent increase in OLE and ARA observed in the transplanted algae, together with the decrease in PALM, were responsible for the significant differences and most of the dissimilarity between T3 and T4 (Table 3).

With regard to FA classes, SFAs, MUFAs and BAFAs showed an increase over time, in contrast to PUFAs, and in particular, the ω-3 class, which displayed an overall opposite pattern (Figure 5). In more detail, PERMANOVA highlighted significant differences for the interactions between time and site for SFAs and ω-3 (SFA: MS = 0.001; Pseudo-F_(1,67)_ = 3.97; *p* < 0.05; ω-3: MS = 0.002; Pseudo-F_(1,67)_ = 2.91; *p* < 0.05, Appendix A): SFA showed a decrease from T0 to T1 followed by an increase in the following times, while ω-3 showed the opposite pattern. At the same time, SFAs and ω-3 FAs also showed differences between treatments and sites (SFA: MS = 0.001; Pseudo-F_(1,67)_ = 6.93; *p* < 0.01; ω-3: MS = 0.004; Pseudo-F_(1,67)_ = 5.84; *p* < 0.05, Appendix A), revealing higher SFA abundance in transplanted than in natural thalli, while, also in this case, ω-3 showed the opposite pattern. MUFAs showed significant differences in the interactions between all three factors (treatment × time × site: MS = 0.002; Pseudo-F_(1,67)_ = 5.56; *p* < 0.05, Appendix A) and, in particular, a significant increase in concentration over time in both natural and transplanted thalli at Barcarello, but not in Capo Gallo. Lastly, the ω-6 class and BAFAs showed significant differences for the time factor (ω-6: MS = 0.008; Pseudo-F_(1,67)_ = 14.70; *p* < 0.05; BAFA: MS = 0.0003; Pseudo-F_(1,67)_ = 71.44; *p* < 0.05, Appendix A) highlighting opposite patterns: ω-6 FAs were significantly more abundant at T0 than at the following times, while BAFAs significantly increased from T0 and T1 to all the subsequent times (Figure 5).

## 4. Discussion

This study revealed a limited stress response to transplantation in the intertidal brown macroalga *Ericaria amentacea* using a combined biochemical approach based on the analysis of total phenolic compounds (TPCs), lipid concentration and fatty acid (FA) composition. Despite biochemical proxies, such as TPCs and FAs, having been used widely to test the response of macroalgae exposed to various abiotic and biotic stressors [32,37,50,51], including through in situ transplantation [33,38,52], to our knowledge, an estimation of the stress induced in macroalgae by experimental handling and transplantation, especially for restoration purposes, has never been carried out.

Overall, the findings of this study for all the biochemical proxies analysed showed similar patterns between natural and transplanted macroalgae over time suggesting that environmental factors and seasonal variability played a significant role in shaping the biochemical profile of *E. amentacea*. Within this framework, the significantly higher TPC and saturated fatty acid (SFAs) concentrations, coupled with the lower lipid and ω-3 fatty acid concentrations recorded in transplanted thalli than in those naturally growing in the same site suggest that a certain degree of modulation of phenolic and lipid metabolism occur under transplantation, consistent with the metabolomic reprogramming ability of macroalgae to mitigate stress effects [33].

Phenolic compounds are a set of secondary metabolites produced by marine macrophytes (algae and seagrasses) with structural, protective and ecological functions [28]. In marine macroalgae, especially intertidal species that are particularly exposed to environmental variability [53], high production and accumulation of TPCs have been observed as a defence function to cope with variations in temperature, salinity, light intensity, depth, pH and other biological factors, such as age, life cycle or herbivore control [33,35,52]. The temporal trend recorded in this study is consistent with these general observations and previous studies on *Cystoseira amentacea* (old name of *E. amentacea*) that reported an increase in TPC concentration from winter to spring and summer, as a defence mechanism to minimize damages from high summer temperature and irradiance in sprouting and growing branches [31,50]. Moreover, brown macroalgae are particularly rich in polyphenols and phlorotannins, which are essential to the physiological integrity of algal tissues, being specifically involved in cell-wall hardening and wound healing [51,54] as well as in long-term acclimation [38]. Although we did not specifically analyse phlorotannins, the higher concentration of TPCs found in transplanted thalli may indicate increased production of phlorotannins as an effect of the manipulative stress induced by transplantation activities.

Similarly, changes in the lipid concentration and fatty acid composition are key molecular mechanisms of many organisms, including macroalgae, for adaptation and acclimation to environmental variations. Here, *E. amentacea* showed overall the typical fatty acid composition of brown macroalgae [55,56,57,58]. Besides the high relative abundance of palmitic acid (16:0, PALM), all samples were rich in oleic acid (18:1 n9, OLE) and ω-3 and ω-6 polyunsaturated fatty acids (PUFAs) with 18 and 20 carbon atoms, namely, arachidonic acid (20:4n-6, ARA) and α-linolenic acid (18:3 n3, ALA) followed by eicosapentaenoic acid (20:5n-3, EPA), stearidonic acid (18:4 n3, SDA) and linoleic acid (18:2 n6, LA). As with TPCs, FA profiles highlighted a similar temporal pattern in natural and transplanted thalli, consisting overall of a decrease in the sum of PUFAs (specifically the ω-3 class) over time, alongside an increase in the ω-6 class, as well as the sum of saturated (SFAs), monounsaturated (MUFAs) and bacterial fatty acids (BAFAs). This pattern reminds the long-term acclimation mechanism that was observed in macroalgae both in the field and laboratory experiments, to cope with rising temperatures based on the adjustment of the cell membrane fatty acids [36,37,59]. The increase in temperature is associated, indeed, with increased fluidity of the cell membrane, which can ultimately cause bilayer disruption [40]. Therefore, the variation in the saturation degree of the membrane FAs resulting from the increase in SFAs over PUFAs as well as the increase in ω-6 over ω-3 and the change of the membrane FA chains (partial substitution of C20 by C18 FAs) have been observed in several macroalgal species to stabilize the cell membranes under high temperatures [18,36,37,39,58,59], maintaining the bilayer liquid crystalline, which is necessary for proper cell functioning [40]. As these processes have been observed in macroalgae of different morpho-functional groups from different latitudes, this biochemical response may represent the winning mechanism by which macroalgae will be able to tolerate the rise in temperature and the increase in marine heatwave frequency forecasted under future ocean warming scenarios, while maintaining proper growth and photosynthetic rates [37]. The increase of BAFAs over time is consistent with the acclimation response of *E. amentacea* to high summer temperatures, as the associated microbiome is also known to change in response to environmental stress [60,61]. On the one side, both the abundance and diversity of the bacterial communities associated with intertidal macroalgae (e.g., *Fucus* spp.) may change when transplanted into the upper intertidal zone, resulting in an increase in thermally tolerant bacteria [60]. On the other side, the abundance of other bacterial strains may also increase under abiotic stress conditions, as the algal microbiome contributes to host resilience by releasing “algal growth and morphogenesis-promoting factors” (AGMPFs) that support normal algal growth and development under stressful conditions [61].

Considering the intrinsic variability of the upper intertidal zone, in terms of temperature, salinity, irradiation and wave action intensity, we cannot distinguish which environmental factor or interaction among such factors may have acted as a stressor contributing to the biochemical changes observed in *E.*
*amentacea* over time. However, against this background, the different levels recorded for ω-3 FAs (higher in transplanted than in natural thalli), ω-6 FAs and SFAs (both higher in natural than in transplanted thalli) can be interpreted as either an amplification of the environmental acclimation response, an increased sensitivity of the transplanted macroalgae to environmental stress or in combination. Consistently, the lower lipid concentration found in transplanted thalli compared to natural ones may depend on a higher sensitivity to oxidative stress and thus to higher levels of lipid peroxidation in the thalli subjected to the mechanical stress of transplantation (e.g., handling, clipping, wounding, prolonged emersion and desiccation, osmotic stress). Membrane lipids are, in fact, the most common targets of lipid peroxidation caused by increased production of reactive oxygen species (ROS) under stressful conditions, including wounding [25,27,62]. In particular, lipid peroxidation ultimately leads to increased free fatty acids [59] and oxygenated PUFA derivatives, such as oxylipins, with known signalling functions in various environmental (biotic and abiotic stress), developmental and immunological responses in macroalgae [63,64]. Therefore, as PUFAs are precursors of oxylipins, the increase in the proportion of ω-6 PUFA to ω-3 PUFAs, is probably due to the need for ω-6 oxylipins, which are stronger than ω-3 oxylipins to accomplish stress adaptation, as already observed in the green macroalga *Ulva lactuca* under heat stress [36].

Overall, the outcome of this manipulative study confirmed the high phenotypic plasticity of intertidal macroalgae as regards biochemical composition [56,65]. Phenotypic plasticity is recognised, indeed, to be crucial in the acclimation and adaptation of macroalgae to the stressful environment of the rocky intertidal zone and increased temperature, e.g., [37,65,66,67]. Here we show that it also helps *E. amentacea* to cope with the stress induced by transplantation activities, confirming the suitability of transplantation of adult thalli for restoration purposes of *E. amentacea* beds. This is also confirmed by previous studies where relatively high survival rates were observed in transplanted adult thalli of *Cystoseira* and *Ericaria* spp. [17,46,68], contrary to what is observed in other brown algae, such as the kelp species *Macrocystis pyrifera* and *Nereocystis luetkeana*, whose survival rates have been found to be rather variable (7–41%) [69,70]. Finally, considering the lack of biochemical stress in transplanted macroalgae, the high mortality rates commonly observed in restoration projects can be attributed to other factors that still need to be explored. In our experiment, at both sites, the survival rate of the transplanted individuals decreased right after the transplantation: while 50% of the thalli transplanted at Barcarello were lost after six weeks, the survival rate of the thalli transplanted in Capo Gallo dropped abruptly just after two weeks, reaching zero before the end of the experiment (Chemello et al., in prep.). In our experiment, a key role is certainly played by the strong hydrodynamics of the rocky intertidal shores, which leads to the detachment of whole thalli (author’s personal observation), while herbivore grazing seems to be more relevant in subtidal areas [71,72].

## 5. Conclusions

*Ericaria amentacea*, both natural and transplanted thalli in an upper intertidal zone, showed clear and comparable temporal trends of total phenolic compound (TPC) content and fatty acid (FA) composition, revealing that, consistent with the literature, seasonal environmental variability, rather than transplantation, had a major effect on their biochemical composition. Increasing TPC concentration, as well as the proportion of saturated FAs (SFAs) over polyunsaturated FAs (PUFAs) and in ω-6 PUFAs over ω-3 PUFAs over the experimental period (spring/summer) in both natural and transplanted *E. amentacea* thalli suggested the occurrence of seasonal acclimation mechanisms to cope with environmental change (i.e., rising temperature, light irradiance, and salinity). In this context, transplant-induced mechanical stress (e.g., manipulation, wounding, temporary drying) seems to have only amplified the adjustments in the biochemical composition of the macroalgae, probably due to a higher sensitivity to environmental and oxidative stress. The high phenotypic plasticity of intertidal species in terms of biochemical composition may contribute towards coping with transplant stress, as already observed for heat stress, confirming that, from the biochemical point of view, transplantation of adult macroalgae thalli from nearby healthy forests seems to be suitable for restoration purposes.

## Figures and Tables

**Figure 1 ijerph-19-12253-f001:**
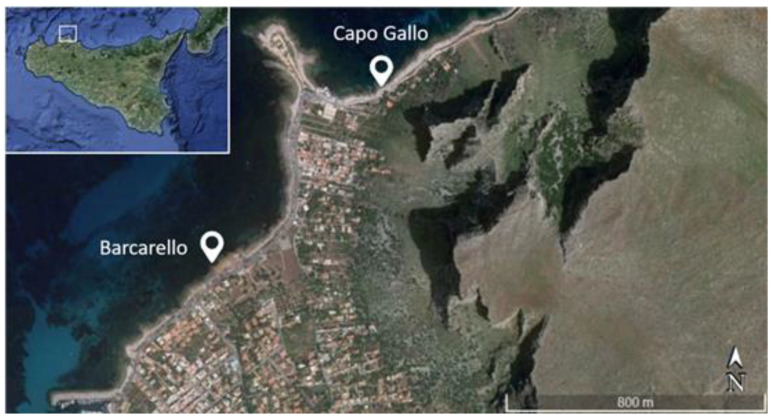
Map of the study area and location of the sites (Barcarello and Capo Gallo) along the northern coast of Sicily (Italy).

**Figure 2 ijerph-19-12253-f002:**
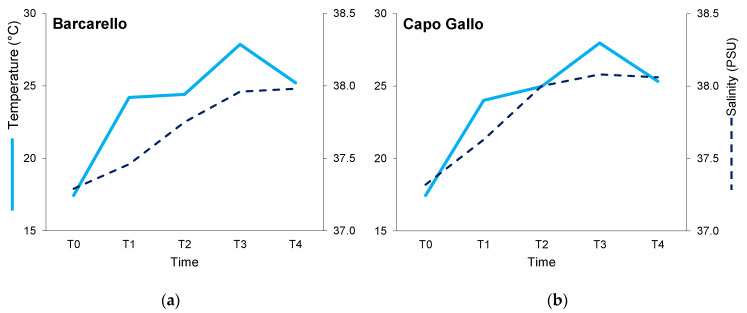
Surface water temperature (°C) and salinity (PSU) measured at the two sites across time: (**a**) Barcarello, (**b**) Capo Gallo.

**Figure 3 ijerph-19-12253-f003:**
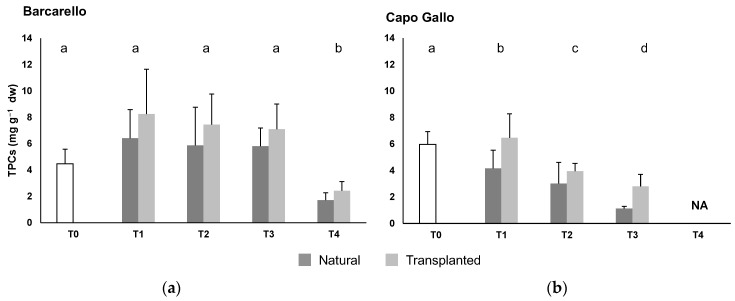
Total phenolic compounds (TPCs; mean ± sd) of *Ericaria amentacea* at the two sites: (**a**) Barcarello; (**b**) Capo Gallo) across time. Based on pair-wise test results, different letters were attributed to significantly different points.

**Figure 4 ijerph-19-12253-f004:**
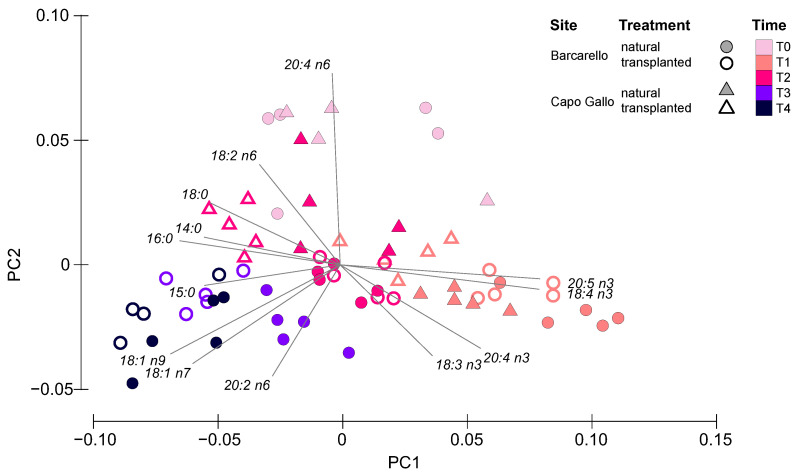
Principal component analysis (PCA) of fatty acid profiles for both natural and transplanted *Ericaria amentacea* thalli throughout the experiment at Barcarello and Capo Gallo sites.

**Figure 5 ijerph-19-12253-f005:**
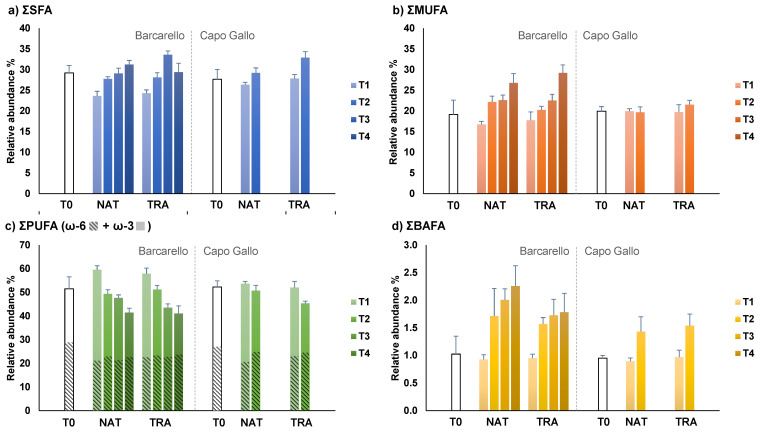
Relative abundance (mean ± standard deviation % of total FAs) of fatty acid classes of *Ericaria amentacea* at T0 and in natural (NAT) and transplanted (TRA) thalli throughout the experiment at Barcarello and Capo Gallo sites. (**a**) Sum of saturated fatty acids SFA; (**b**) sum of monounsaturated fatty acids MUFAs; (**c**) sum of polyunsaturated fatty acids PUFA with the distinction between ω-6 and ω-3; (**d**) sum of bacterial fatty acids BAFAs.

**Table 1 ijerph-19-12253-t001:** Fatty acid relative abundance (mean ± standard deviation % of total FAs) of natural and transplanted *Ericaria amentacea* thalli, throughout the experiment at the Barcarello site. The concentration of total lipids (TL mg g^−1^) is also reported. The sums of the different fatty acid categories and total lipids are highlighted in bold.

Site Barcarello			Natural								Transplanted							
Time	T0		T1		T2		T3		T4		T1		T2		T3		T4	
**FAs**	mean	sd	mean	sd	mean	sd	mean	sd	mean	sd	mean	sd	mean	sd	mean	sd	mean	sd
14:0	4.0	0.4	3.0	0.3	3.5	0.7	3.6	0.2	4.2	0.4	3.0	0.2	3.4	0.4	4.0	0.3	4.1	0.2
15:0	0.2	0.0	0.2	0.0	0.2	0.1	0.2	0.1	0.3	0.0	0.1	0.0	0.2	0.0	0.2	0.0	0.3	0.0
16:0 (PALM)	23.4	1.8	19.6	1.1	22.9	0.9	24.1	1.4	25.4	1.0	20.3	0.9	23.0	0.9	27.8	0.6	23.9	2.1
17:0	0.3	0.0	0.0	0.0	0.0	0.0	0.0	0.0	0.0	0.0	0.0	0.0	0.0	0.0	0.0	0.0	0.0	0.0
18:0	0.5	0.1	0.3	0.1	0.4	0.1	0.5	0.1	0.4	0.0	0.3	0.1	0.5	0.0	0.8	0.1	0.4	0.0
20:0	0.1	0.0	0.1	0.0	0.0	0.0	0.1	0.1	0.1	0.1	0.0	0.0	0.1	0.0	0.1	0.1	0.1	0.0
LCFAs (>22:0)	0.5	0.1	0.4	0.1	0.6	0.2	0.6	0.1	0.6	0.1	0.4	0.1	0.5	0.1	0.5	0.1	0.5	0.1
**Ʃ** **SFA**	**29.2**	**1.8**	**23.7**	**1.1**	**27.8**	**0.5**	**29.1**	**1.3**	**31.2**	**1.0**	**24.3**	**0.8**	**28.1**	**1.1**	**33.6**	**0.9**	**29.4**	**2.1**
16:1 n7 (PALMo)	5.0	1.2	4.2	0.5	5.6	1.4	4.4	0.5	6.3	0.5	4.0	0.5	4.5	0.3	4.5	0.5	6.2	0.6
18:1 n7	0.9	0.3	0.9	0.1	1.1	0.1	1.6	0.1	1.8	0.2	0.9	0.1	1.2	0.1	1.4	0.2	1.6	0.3
18:1 n9 (OLE)	13.2	2.1	11.5	0.6	15.5	0.4	16.6	1.1	18.5	2.5	12.8	1.6	14.5	0.7	16.6	1.5	21.4	2.2
**Ʃ** **MUFA**	**19.2**	**3.5**	**16.7**	**0.7**	**22.2**	**1.4**	**22.6**	**1.2**	**26.8**	**2.2**	**17.8**	**2.0**	**20.3**	**0.9**	**22.5**	**1.5**	**29.2**	**1.9**
18:2 n6 (LA)	3.9	0.9	2.7	0.3	3.0	0.5	3.7	0.5	3.6	0.6	2.7	0.5	3.3	0.4	4.6	0.3	2.5	0.4
18:3 n3 (ALA)	9.9	0.8	13.2	1.3	12.1	1.3	12.6	1.0	10.8	0.6	12.8	1.8	12.8	0.8	11.4	0.5	10.3	0.5
18:3 n6	0.4	0.1	0.4	0.1	0.3	0.1	0.2	0.0	0.3	0.0	0.3	0.1	0.3	0.0	0.3	0.2	0.2	0.0
18:4 n3 (SDA)	4.5	1.9	10.8	1.7	5.1	0.7	3.0	1.1	2.1	0.6	9.3	0.5	5.0	0.6	2.4	0.2	1.6	0.7
20:2 n6	0.5	0.2	0.6	0.0	0.8	0.3	0.8	0.2	1.2	0.2	0.7	0.2	0.6	0.2	0.4	0.1	1.3	0.2
20:3 n3	0.2	0.1	0.3	0.1	0.3	0.1	0.3	0.1	0.3	0.1	0.4	0.1	0.4	0.1	0.1	0.1	0.3	0.1
20:3 n6	1.4	0.3	1.0	0.2	0.9	0.2	0.6	0.2	1.8	0.7	1.0	0.2	1.1	0.2	1.2	0.2	2.0	0.5
20:4 n3	1.1	0.1	2.0	0.2	1.7	0.1	1.7	0.1	1.3	0.2	2.0	0.2	1.8	0.2	1.5	0.2	1.4	0.1
20:4 n6 (ARA)	22.2	1.4	16.1	0.7	17.7	0.9	15.8	0.8	15.3	0.5	17.5	0.6	17.5	0.6	15.9	0.8	17.2	2.5
20:5 n3 (EPA)	7.0	1.6	12.1	0.9	7.5	0.8	7.9	0.6	4.3	0.7	11.1	0.7	8.0	0.8	4.6	0.8	3.8	0.9
22:4 n6	0.1	0.0	0.2	0.0	0.2	0.1	0.2	0.1	0.3	0.1	0.1	0.0	0.1	0.0	0.1	0.1	0.2	0.0
22:6 n3	0.0	0.0	0.1	0.0	0.0	0.0	0.9	0.3	0.0	0.0	0.0	0.0	0.1	0.1	0.8	0.9	0.0	0.0
**Ʃ** **ω-3**	**22.8**	**4.0**	**38.6**	**2.3**	**26.6**	**1.9**	**26.4**	**1.5**	**18.9**	**1.8**	**35.5**	**2.1**	**28.1**	**1.9**	**20.9**	**1.8**	**17.4**	**1.8**
**Ʃ** **ω-6**	**28.7**	**2.5**	**21.0**	**1.0**	**22.8**	**0.6**	**21.3**	**0.6**	**22.5**	**0.2**	**22.3**	**0.5**	**23.1**	**0.9**	**22.6**	**0.4**	**23.6**	**3.1**
**Ʃ** **PUFA**	**51.5**	**5.1**	**59.6**	**1.6**	**49.4**	**1.7**	**47.7**	**1.2**	**41.4**	**1.9**	**57.9**	**2.4**	**51.2**	**1.7**	**43.5**	**1.6**	**41.1**	**3.2**
**Ʃ** **PUFA/** **Ʃ** **SFA**	**1.8**	**0.3**	**2.5**	**0.2**	**1.8**	**0.1**	**1.6**	**0.1**	**1.3**	**0.1**	**2.4**	**0.2**	**1.8**	**0.1**	**1.3**	**0.1**	**1.4**	**0.2**
Anteiso	0.1	0.0	0.0	0.0	0.3	0.2	0.2	0.0	0.3	0.1	0.0	0.0	0.2	0.0	0.1	0.1	0.2	0.0
-OH	0.0	0.0	0.0	0.0	0.2	0.2	0.2	0.1	0.1	0.1	0.0	0.0	0.2	0.1	0.1	0.1	0.0	0.1
**Ʃ** **BAFA ***	**1.0**	**0.3**	**0.9**	**0.1**	**1.7**	**0.5**	**2.0**	**0.2**	**2.3**	**0.4**	**0.9**	**0.1**	**1.6**	**0.1**	**1.7**	**0.3**	**1.8**	**0.3**
**TL (mg g^−1^)**	**38.4**	**4.2**	**33.2**	**5.2**	**30.2**	**1.7**	**28.4**	**2.6**	**26.7**	**2.5**	**29.2**	**3.2**	**27.6**	**2.3**	**28.2**	**3.2**	**24.3**	**5.4**

FAs ≤ 0.1% in all samples were omitted. SFA: saturated FA; MUFA: monounsaturated FA; PUFA: polyunsaturated FA; BAFA: bacterial FA; LCFA: long-chain FA; PALM: palmitic acid; PALMo: palmitoleic acid; OLE: oleic acid; LA: linoleic acid, ALA: α-linolenic acid; SDA: stearidonic acid; ARA: arachidonic acid, EPA: eicosapentaenoic acid. * ΣBAFA is the sum of branched FAs, -OH hydroxyl FAs and 18:1 n7.

**Table 2 ijerph-19-12253-t002:** Fatty acid relative abundance (mean ± standard deviation % of total FAs) of natural and transplanted *Ericaria amentacea* thalli, throughout the experiment at the Capo Gallo site. The concentration of total lipids (TL mg g^−1^) is also reported. The sums of the different fatty acid categories and total lipids are highlighted in bold.

Site Capo Gallo			Natural				Transplanted			
Time	T0		T1		T2		T1		T2	
**FAs**	mean	sd	mean	sd	mean	sd	mean	sd	mean	sd
14:00	3.6	0.4	2.1	0.1	3.4	0.2	3.1	0.3	3.7	0.7
15:00	0.2	0.0	0.1	0.0	0.2	0.0	0.2	0.0	0.2	0.1
16:0 (PALM)	22.7	2.0	23.5	0.6	24.4	1.0	23.5	0.8	27.6	1.6
18:00	0.4	0.1	0.1	0.0	0.5	0.1	0.3	0.0	0.6	0.1
20:00	0.2	0.0	0.0	0.0	0.1	0.0	0.1	0.0	0.1	0.1
LCFAs (>22:0)	0.4	0.1	0.3	0.1	0.5	0.1	0.5	0.1	0.5	0.2
**Ʃ** **SFA**	**27.7**	**2.4**	**26.3**	**0.6**	**29.2**	**1.2**	**27.8**	**1.0**	**32.9**	**1.4**
16:1 n7 (PALMo)	6.0	0.6	5.4	0.6	5.0	0.5	5.1	0.5	4.9	0.2
18:1 n7	0.9	0.0	0.9	0.1	1.1	0.2	1.0	0.1	1.3	0.2
18:1 n9 (OLE)	13.1	1.2	13.6	0.6	13.6	1.4	13.6	1.2	15.2	1.0
**Ʃ** **MUFA**	**20.0**	**1.1**	**20.0**	**0.6**	**19.7**	**1.3**	**19.7**	**1.8**	**21.5**	**1.0**
18:2 n6 (LA)	4.1	0.5	2.5	0.2	3.8	0.5	3.1	0.4	4.4	0.9
18:3 n3 (ALA)	9.3	0.6	10.7	0.6	11.8	1.2	11.0	1.0	10.1	0.7
18:3 n6	0.4	0.1	0.1	0.1	0.3	0.1	0.3	0.1	0.3	0.1
18:4 n3 (SDA)	5.8	3.3	9.1	1.2	4.2	1.1	6.5	1.0	2.9	0.5
20:2 n6	0.5	0.2	0.6	0.1	0.4	0.2	0.6	0.2	0.3	0.2
20:3 n3	0.3	0.1	0.2	0.1	0.2	0.0	0.3	0.1	0.1	0.1
20:3 n6	1.2	0.3	0.9	0.1	1.0	0.2	1.0	0.3	1.1	0.1
20:4 n3	1.4	0.1	2.1	0.2	1.7	0.1	1.9	0.2	1.5	0.1
20:4 n6 (ARA)	20.6	3.0	16.3	0.5	19.3	2.0	17.8	0.7	18.1	0.9
20:5 n3 (EPA)	8.4	1.9	11.0	0.9	7.8	1.0	9.3	0.8	5.9	0.9
22:4 n6	0.0	0.0	0.1	0.0	0.1	0.1	0.2	0.1	0.2	0.0
22:6 n3	0.0	0.0	0.0	0.0	0.1	0.0	0.0	0.0	0.2	0.2
**Ʃ** **ω-3**	**25.2**	**5.7**	**33.2**	**1.4**	**25.8**	**3.1**	**29.1**	**2.4**	**20.8**	**1.1**
**Ʃ** **ω-6**	**27.0**	**3.5**	**20.5**	**0.5**	**24.9**	**2.4**	**22.9**	**1.0**	**24.5**	**0.9**
**Ʃ** **PUFA**	**52.2**	**2.6**	**53.6**	**1.0**	**50.7**	**2.2**	**52.0**	**2.6**	**45.3**	**1.0**
**Ʃ** **PUFA/** **Ʃ** **SFA**	**1.9**	**0.3**	**2.0**	**0.1**	**1.7**	**0.1**	**1.9**	**0.2**	**1.4**	**0.1**
Anteiso	0.1	0.0	0.0	0.0	0.1	0.0	0.0	0.0	0.2	0.0
-OH	0.0	0.0	0.0	0.0	0.1	0.1	0.0	0.0	0.0	0.0
**Ʃ** **BAFA ***	**0.9**	**0.1**	**0.9**	**0.1**	**1.4**	**0.3**	**1.0**	**0.1**	**1.5**	**0.2**
**TL (mg g^−1^)**	**37.9**	**2.6**	**38.1**	**4.2**	**39.2**	**12.4**	**27.7**	**5.7**	**29.2**	**8.0**

FAs ≤ 0.1% in all samples were omitted. The meaning of the acronyms is the same as in Table 1 *.

**Table 3 ijerph-19-12253-t003:** SIMPER analysis showing fatty acids of *Ericaria amentacea* thalli that contribute most to the dissimilarity between times at each site and treatment.

Comparison between Time	Site	Natural			Transplanted		
		FA	Contrib%	Cum%	FA	Contrib%	Cum%
T0 vs. T1	Barcarello	SDA	18.6	18.6	SDA	16.5	16.5
		ARA	18.2	36.8	ARA	16.5	33.0
		EPA	15.0	51.7	EPA	13.7	46.7
	Capo Gallo	ARA	20.2	20.2	ARA	19.4	19.4
		SDA	16.2	36.4	SDA	17.3	36.7
		EPA	10.9	47.3	PALM	10.7	47.3
T1 vs. T2	Barcarello	SDA	22.1	22.1	SDA	23.5	23.5
		EPA	17.9	40.0	EPA	16.6	40.0
		OLE	15.1	55.1	PALM	14.8	54.8
	Capo Gallo	SDA	24.2	24.2	PALM	20.3	20.3
		EPA	15.7	39.9	SDA	17.7	38.0
		ARA	14.7	54.6	EPA	16.7	54.7
T2 vs. T3	Barcarello	SDA	14.6	14.6	PALM	22.1	22.1
		ARA	12.8	27.4	EPA	15.6	37.7
		PALM	10.1	37.5	SDA	12.0	49.7
T3 vs. T4	Barcarello	EPA	19.3	19.3	OLE	20.1	20.1
		OLE	12.7	32.0	PALM	19.4	39.5
		PALMo	9.9	41.9	ARA	10.6	50.1

Contrib%: dissimilarity contribution. Cum%: dissimilarity cumulative contribution. The meanings of the acronyms are the same as in Table 1.

## Data Availability

The data presented in this study are available in tables, figures, and Appendix A.

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
