# Peer review of "Limited Stress Response to Transplantation in the Mediterranean Macroalga Ericaria amentacea, a Key Species for Marine Forest Restoration"

_ijerph, 2022, doi:10.3390/ijerph191912253_

Round 1
Reviewer 1 Report
This paper presents an interesting topic of potential stress caused by the handling of the thalli of Ericaria amentacea. Increasing anthropic pressure, but also climate changes, contribute to major changes in the ecosystems, and today we are in the phase of making decisions as quickly as possible to restore and renature the affected areas. Therefore, the study presents a subject of major interest with useful information both for the academic forum and for the general public, but especially for the rangers of protected areas. The approach and description is at a high level of European research in the last decade and responds to the needs of further development of knowledge about the to assess the stress response to transplantation of the macroalga Ericaria amentacea.
The subject is well documented in Introduction. The data presented by the authors are original and significant. The study is correctly designed. In general, the statistical analyses are performed with good technical standards.
Author Response
Dear Reviewer, thank you very much for your comments
Reviewer 2 Report
The paper is well written, and it describes the study in a clear and easy-to-read way. The results consistently answer the research questions, and the conclusions that the authors extract are sound. Therefore, I would recommend the paper for publication after some minor changes I've included as comments in the PDF.

Author Response
Dear Reviewer, thank you very much for your comments
We made all the suggested changes. Below are the point-by-point responses (R) to the comments (C) made in the PDF:
C: Line 42: Why does reference #8 appear before #7?
R: There was an error in the listed references. We have now corrected it and are listed in the right order (ines 40-43).
C: Lines 120-121: Please revise writing. It could be simplified and therefore clarified
R: The whole paragraph related to the transplant protocol was rephrased to improve readability (lines 118-127). Moreover, there was an error in the references used for the transplantation protocol (39 and 40, instead of 45 and 46). We have now corrected them.
C: Line 126: I haven't found the reason why epoxy putty is used in the process of transplanting in any of the references you provide for the protocol. Can you explain what is the role/importance of using this substance in this case?
R: We have added the reason why using epoxy putty and the related references as follows: “… using epoxy putty (Stucchi Veneziani) to fix the thalli to the substrate [45,46]” (lines 126-127).
C: Line 415: From the perspective of restorationists, it could be interesting to compare this survival rate with data from other species (other algae used for restoration, or even plants used in land restoration projects), to provide some context
R: A comparison with the survival rate of other brown algae was added in the previous sentence where there was already mention of the survival rate of Cystoseira and Ericaria spp. as follows: "Here we show that it also helps E. amentacea to cope with the stress induced by transplantation activities, confirming the suitability of transplantation of adult thalli for restoration purposes of E. amentacea beds. This is also confirmed by previous studies where relatively high survival rates were observed in transplanted adult thalli of Cystoseira and Ericaria spp. [17,46,69], contrary to what is observed in other brown algae such as the kelp species Macrocystis pyrifera and Nereocystis luetkeana whose survival rates have been found to be rather variable (7 – 41%) [70,71]". (lines 410-412).